# Erythrocyte sedimentation rate and hemoglobin-binding protein in free-living box turtles (*Terrapene* spp.)

**Laura Adamovicz**[1]*, **Sarah J. Baker**[1,2], **Ethan Kessler**[1,3], **Marta Kelly**[1], **Samantha Johnson**[1], **John Winter**[1], **Christopher A. Phillips**[3], **Matthew C. Allender**[1]

**1** Wildlife Epidemiology Laboratory, University of Illinois, Urbana, Illinois, United States of America, **2** Arizona Game and Fish Department, Phoenix, Arizona, United States of America, **3** Illinois Natural History Survey Prairie Research Institute, Champaign, Illinois, United States of America

* adamovi2@illinois.edu

**Data Availability Statement:** All relevant data are within the manuscript and its Supporting Information files.

## Abstract

The acute phase response is a highly conserved reaction to infection, inflammation, trauma, stress, and neoplasia. Acute phase assays are useful for wildlife health assessment, however, they are infrequently utilized in reptiles. This study evaluated erythrocyte sedimentation rate (ESR) in eastern (*Terrapene carolina carolina*) and ornate box turtles (*Terrapene ornata ornata*) and hemoglobin-binding protein (HBP) in *T. ornata*. Erythrocyte sedimentation rate in 90 *T. carolina* and 105 *T. ornata* was negatively associated with packed cell volume and was greater in unhealthy turtles ($p < 0.05$). Female *T. ornata* had higher ESR values than males ($p < 0.05$). Measurement of ESR with a microhematocrit tube proportionally overestimated values from a commercial kit (Winpette), though both methods may retain utility with separate reference intervals. Hemoglobin-binding protein concentration in 184 *T. ornata* was significantly increased in adults and unhealthy turtles ($p < 0.05$). Erythrocyte sedimentation rate values were similar between seasons and populations, and HBP values were consistent between years, indicating that these analytes may have more stable baseline values than traditional health metrics in reptiles. This study demonstrates that ESR and HBP are promising diagnostics for health assessment in wild box turtles. Incorporating these tests into wild herptile health assessment protocols may support conservation efforts and improve ecosystem health monitoring.

## Introduction

The acute phase response is a series of highly conserved transcriptomic, proteomic, and metabolomic reactions to infection, trauma, neoplasia, inflammation, and physiologic stress [1,2]. Acute phase assays are commonly used to identify subclinical disease, monitor the progression of inflammatory processes, and aid prognostication [1–3]. Some acute phase biomarkers are highly sensitive for detecting pathologic changes and are valuable for differentiating healthy and unhealthy animals [1,4–8]. Many of these assays can be interpreted at both the individual

**Funding:** Our research was partially funded by the Illinois Department of Natural Resources (State Wildlife Grant T104-R-1; MCA, LA), which supported field work and sampling supplies. Website: https://www.dnr.illinois.gov/conservation/IWAP/Pages/StateWildlifeGrants.aspx We were also partially funded by the Friends of Nachusa, which supported ornate box turtle erythrocyte sedimentation rates and hemoglobin binding protein kits (MCA, LA). Website: https://www.nachusagrasslands.org/ The funders had no role in study design, data collection and analysis, decision to publish, or preparation of the manuscript.

**Competing interests:** The authors have declared that no competing interests exist.

and population levels and do not require species-specific reagents, making them useful for wildlife health studies [9–15].

Assessing wildlife health status can be challenging due to poor antemortem recognition of disease, limited understanding of pathogen epidemiology, and a lack of validated diagnostic tests [16,17]. Exacerbating these issues, wild reptiles display significant physiologic variation in clinical pathology values based on age, sex, season, and reproductive state [18–25]. For these reasons, routine diagnostic tests such as hematology and plasma biochemistries frequently have poor discriminatory power for identifying unhealthy reptiles [25]. Improving reptile health assessment is important to advance veterinary practice, support effective conservation strategies, and improve ecosystem health monitoring using reptilian sentinels [26]. Acute phase response testing, which can sensitively and non-specifically screen for evidence of underlying pathology, may augment reptile health assessments and facilitate the identification of individuals and populations in need of intervention [14,15].

Eastern (*Terrapene carolina carolina*) and ornate box turtles (*Terrapene ornata ornata*) are biosentinel species in decline due to habitat destruction and fragmentation, road mortality, overcollection for human use, and predation [27,28]. Box turtles are also increasingly threatened by infectious diseases (e.g. ranavirus, *Mycoplasma* sp., herpesviruses, adenovirus) and toxicants (e.g. organochlorines, heavy metals); underscoring the need for reliable tools to characterize emerging health threats [29–40]. Acute phase response testing in box turtles may supplement existing diagnostic modalities and improve health assessment protocols; ultimately supporting conservation goals and enhancing the evaluation of ecosystem wellness [41].

Haptoglobin is a positive acute phase protein that scavenges free hemoglobin to prevent oxidative damage and inhibit bacterial growth [1,2]. Avian and reptile haptoglobin analogs are referred to as "hemoglobin-binding proteins" (HBP) to reflect the potential for an alternative genetic origin [42]. HBP quantification uses commercially available colorimetric kits that detect the binding of HBP to hemoglobin, and previous studies suggest that the results are reliable across species [1,43,44]. HBP concentrations have been reported in free-living *T. carolina* [23], but this diagnostic tool has not yet been explored in *T. ornata*.

Erythrocyte sedimentation rate (ESR) is an indirect acute phase analyte measuring the distance that erythrocytes settle out of plasma after one hour. Greater ESR values are associated with a variety of disease states in humans, supporting the use of this test as a non-specific indicator of overall health status [45–47]. ESR has superior discriminatory power for differentiating healthy and unhealthy gopher tortoises (*Gopherus polyphemus*) compared to hematology, fibrinogen, protein electrophoresis, and lactate, making this a promising diagnostic for chelonian health assessment [48]. While ESR is clinically useful for health assessment in many taxa, including chelonians, commercially-available ESR kits require a relatively large blood volume (0.6 – 2mL) which may prohibit their use in small species and juveniles. To address this logistical challenge, the present study investigated an alternative method for ESR measurement using microhematocrit tubes [3,48–52].

This study had four main objectives. First, to assess agreement between three ESR measurement methods in free-living box turtles. Second, to generate reference intervals for ESR and HBP. Third, to characterize demographic (sex, age class), physiologic (packed cell volume), and spatiotemporal (season, study site) associations with ESR and HBP values. Fourth, to investigate the association between ESR, HBP, and health status. Based on previous studies, we hypothesized that there would be good agreement between the three ESR measurement methods [53–55], ESR would be negatively associated with packed cell volume (PCV) [45,46], and both ESR and HBP would elevate in adults [10,23,49,56–59], females [23,59–64], and turtles with clinical signs of illness or injury [15,48,65–67].

## Materials and methods

### Fieldwork

Ornate box turtles were evaluated at two public sites in Lee County, Illinois during May 2016, 2017, and 2018. Eastern box turtles were evaluated at four different capture sites in Vermilion County, Illinois in mid-May (spring sampling) and early August (summer sampling) 2018. Exact locations of field sites are available upon request.

Turtles were located using a combination of human and canine searches [68]. Each animal was weighed to the nearest gram, assigned to an age class (> 200 g was considered an adult, < 200 g was considered a juvenile), and sexed using a combination of plastron shape, eye color, and tail length [69].

Complete physical examinations were performed by a single observer (LA). Turtles displaying clinical signs of illness (ocular/nasal discharge, oral plaques, open-mouth breathing, etc.) or active injuries were categorized as "unhealthy", while turtles with no clinical signs of illness or fully-healed injuries were considered "apparently healthy". Blood samples (< 0.8% body weight) were collected from the subcarapacial sinus using a 1.5" 22 gauge needle and a 3cc syringe. If obvious lymph contamination of the sample was observed, it was discarded and a new sample was collected. Blood samples were immediately placed into lithium heparin microtainers (Becton Dickinson Co., Franklin Lakes, NJ 07417) for ESR and packed cell volume determination and lithium heparin plasma separator tubes (Becton Dickinson Co.) for HBP assays. Blood samples were stored on wet ice until processing (1–3 hours). Turtles were permanently identified by marginal scute notching as previously described [70]. Briefly, square notches (< 3 mm x 3 mm) were placed in the center of up to four marginal scutes using a hand-held hacksaw blade disinfected between turtles with 2% chlorhexidine [70]. Turtles were then released at their original sites of capture. Protocols for animal handling and sampling were approved by the University of Illinois Institutional Animal Care and Use Committee (#15017 and 18000).

### Clinical pathology

Packed cell volume (PCV) was determined using sodium heparinized microhematocrit tubes (Jorgensen Laboratories,Inc., Loveland, CO 80538) centrifuged at 14,500 rpm for five minutes.

Erythrocyte sedimentation rate was performed with a commercially available kit using the Wintrobe method (Winpette, Guest Scientific AG, Switzerland). Briefly, heparinized blood tubes were inverted at least 8 times to ensure even mixing, the provided reservoir tube was filled with 0.6mL of heparinized blood, the Winpette was inserted, and the entire unit was placed into a leveled ESR stand (Guest Scientific AG, Switzerland). The result was read at 60 minutes using 1) the millimeter markings on the Winpette and 2) digital calipers with sub-millimeter precision (Digi-max Caliper with LCD Readout, SP Scienceware, Wayne, NJ 07470).

For comparison, ESR was also performed using a microhematocrit tube. This method is cost-effective, readily available, and requires < 0.1mL of blood, making it attractive for application to small wildlife species. Furthermore, microhematocrit tube measurements have proven to correlate well with gold standard ESR methodologies in people [53–55]. The microhematocrit tube was filled approximately ¾ full, plugged with clay at one end, and positioned upright in a leveled stand (Critoseal Identi-Tray, Leica Microsystems, Wetzlar, Germany). The result was read at 60 minutes using digital calipers. The Winpette and microhematocrit tube assays were initiated within 2 minutes of each other for each blood sample to prevent bias due to differences in storage time. Intra-assay coefficients of variation were determined for approximately 10% of the study population. Inter-assay ESR assessments require commercially-

available quality control materials, and were not pursued [46,47]. Samples with air bubbles within the columns were excluded from analysis for both methods.

Plasma was separated from red blood cells within 3 hours of collection via centrifugation at 6,000 rpm for 10 minutes (Clinical 200 Centrifuge, VWR, Radnor, PA 19087, USA). Plasma aliquots were frozen at -80°C from 2 months to 2 years to allow batching of HBP tests using a commercially available kit (Haptoglobin Phase Colorimetric Assay, Tridelta Development Ltd., Maynooth, Ireland). Plasma samples and standards were assayed in duplicate and results were read at 630nm (Synergy 2 Multi-Mode Microplate Reader, BioTek Instruments, Inc, Winooski, VT 05404). Absorbance values were averaged for each sample, and HBP was quantified based on a five-point standard curve from 0–2.5 mg/mL. To assess linearity of HBP quantitation, pooled plasma was diluted 0%, 20%, 40%, 60%, 80%, and 100% with sterile saline and a linear regression model was applied to the results. Passing-Bablok agreement analysis was performed between observed HBP values and those predicted by the linear regression model. A runs test was applied to determine whether the observed data differed significantly from the linear model. Intra-assay and inter-assay coefficients of variation (CV) were determined, and the limit of detection was set as the mean concentration of blank samples, as determined by the standard curve.

Plasma within PCV tubes, Winpettes, and plasma separator tubes was visually assessed for lipemia and hemolysis. If present, samples were excluded from analysis.

## Statistical analyses

All statistical assessments were performed using R version 3.5.1 at an alpha value of 0.05 [71]. Data distributions were assessed for normality using histograms, skewness, kurtosis, and the Shapiro-Wilk statistic. Summary data including means, standard deviations, and ranges (normally distributed data); medians, $10^{th}$ and $90^{th}$ percentiles, and ranges (non-normally distributed data); and counts (categorical data) were tabulated.

Differences in categorical variables (sex, age class) between study sites were evaluated using Fisher's exact tests. Sex ratios were evaluated using binomial tests (expected ratio 0.5). Differences in PCV between study sites, seasons, and sexes were assessed using general linear models. The effects of PCV, study site, season, sex, age class, and health classification ("unhealthy" vs. "apparently healthy") on ESR and HBP were assessed using general linear models. Casewise deletion was performed for turtles with missing data, then sets of candidate models predicting ESR and HBP were ranked using information-theoretic approaches (Akaike's information criterion; AIC) with the AICcmodavg package [72].

Reference intervals were constructed using data from "apparently healthy" turtles according to American Society for Veterinary Clinical Pathology guidelines [73]. Outliers were visually identified using box plots and excluded using Horn's method [74]. The nonparametric method was used to generate 95% reference intervals for ESR and HBP based on the observations of Le Boedec et al. [75]. Ninety percent confidence intervals were generated around the upper and lower bounds of each reference interval using nonparametric bootstrapping with 5000 replicates. The width of the confidence intervals (WCI) was compared to the total width of the reference interval (WRI) to infer the need for a larger sample size (improved n is recommended when WCI/WRI > 0.2). All reference interval generation was performed using the referenceIntervals package [76].

Agreement in ESR values from the different measurement methods was evaluated using Passing-Bablok regression and Bland-Altman plots (packages mcr, BlandAltmanLeh) [77–80].

## Results

### Assay performance

The inter-assay CV for the HBP assay was 4.23%, (95% CI: 1.68–6.79%) while the intra-assay CV was 3.82%, (95% CI: 2.23–5.41%). Linearity of HBP detection under dilution was supported by Passing-Bablok regression because the 95% confidence interval of the slope contained 1 (0.77–1.18) and the 95% confidence interval for the y-intercept contained 0 (-0.04–0.05). The runs test also supported linearity with a non-significant p-value (p = 0.648). The limit of detection was 0.01 mg/mL.

The intra-assay CV for Winpette ESR was 7.9% (95% CI: 1–15%), while for the Winpette caliper method it was 7.2% (95% CI: 2–12%). The intra-assay CV for microhematocrit tube ESR was lower at 3.2% (95% CI: 2–4.5%).

### Erythrocyte sedimentation rate

Erythrocyte sedimentation rate was performed in 90 *T. carolina* and 105 *T. ornata* from all study sites in 2018 (Table 1, Fig 1). Six *T. carolina* were classified as "unhealthy" due to active shell injuries (N = 4), peeling, discolored scutes (N = 1), a draining pedal granuloma (N = 1), and the presence of oral ulcers (N = 1). Eight *T. ornata* were classified as "unhealthy" due to active shell injuries.

A significant male bias was observed during the spring sampling period for *T. carolina* (p = 0.002). Male turtles of both species had higher PCV values than females (*T. carolina* effect size = 3.75%, p = 0.02; *T. ornata* effect size = 2.1%, p = 0.001), and PCV was higher in the summer compared to the spring (*T. carolina* effect size = 3.4%, p = 0.03). Due to potential confounding, both season (*T. carolina* only) and sex were included in models evaluating the effects of PCV on ESR.

Erythrocyte sedimentation rate was negatively associated with PCV and was greater in unhealthy *T. carolina* and *T. ornata* (Table 2). Female *T. ornata* had greater ESR values than males. Season, age class, and study site were not significantly associated with ESR (p > 0.05).

The most parsimonious models for ESR in *T. carolina* contained the additive effects of season, sex, PCV, and health classification (S1 Table). Top models had adjusted $R^2$ values of 0.33–0.44 and p-values < 0.0001. The most parsimonious models for ESR in *T. ornata* contained the additive effects of sex, PCV, and health classification (adjusted $R^2$ = 0.17–0.22, p < 0.0001; S2 Table).

Bland-Altman plots and Passing-Bablok analyses revealed similar results for agreement between the three evaluated ESR methods. In *T. carolina*, the microhematocrit tube method

**Table 1. Population demographics of free-living eastern (*Terrapene carolina carolina*) and ornate box turtles (*Terrapene ornata ornata*) sampled for erythrocyte sedimentation rate (ESR) during 2018 and Hemoglobin-Binding Protein (HBP) over three years.**

| | Erythrocyte Sedimentation Rate | | | Hemoglobin-binding Protein | | |
|---|---|---|---|---|---|---|
| | *T. carolina* | | *T. ornata* | *T. ornate* | | |
| | Spring | Summer | Spring | 2016 | 2017 | 2018 |
| **Sex** | | | | | | |
| Female | 18 | 14 | 46 | 12 | 28 | 25 |
| Male | 40 | 18 | 59 | 29 | 49 | 30 |
| Unknown | 0 | 0 | 0 | 0 | 10 | 1 |
| **Age Class** | | | | | | |
| Adult | 57 | 32 | 101 | 41 | 78 | 52 |
| Juvenile | 1 | 0 | 4 | 0 | 0 | 13 |

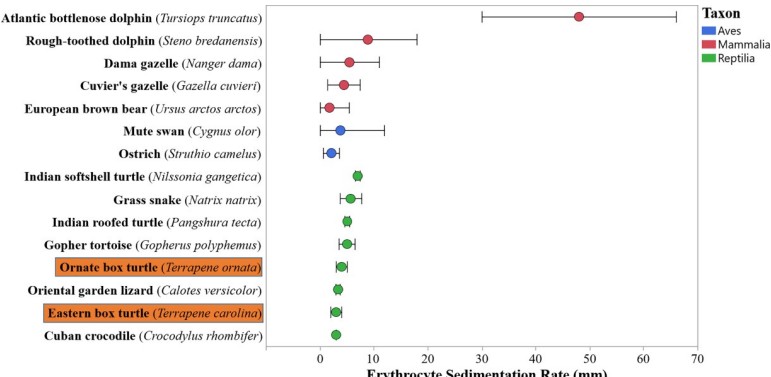

**Fig 1. Comparison of erythrocyte sedimentation rate in free-living and zoo-maintained wildlife from the present study (eastern box turtles, *Terrapene carolina carolina* and ornate box turtles, *Terrapene ornata ornata*) and the literature [25,52,56,60–63,81–85].** Circles represent measures of central tendency (mean or median) and bars represent a measure of dispersion (standard deviation or 25th and 7th percentiles).

proportionally overestimated ESR compared to both the Winpette and Winpette caliper methods (S1 & S2 Figs, Table 3). In *T. ornata*, microhematocrit tubes proportionally overestimated the Winpette caliper method, but not the Winpette method (S1 & S2 Figs, Table 3).

Reference intervals for each ESR measurement method in each species are reported in Table 4. Results for the juvenile *T. carolina* were 4 mm (Winpette), 4.1 mm (Winpette calipers), and 6 mm (microhematocrit tube). Values for the four juvenile *T. ornata* included 2, 4, 4, 6 mm (Winpette), 3.1, 3.3, 3.8, 5.7 mm (Winpette calipers) and 4.1, 4.3, 4.9, 6.6 mm (microhematocrit tube). The ESR reference interval boundaries for the Winpette and Winpette caliper methods in *T. ornata* were somewhat imprecise (WCI/WRI > 0.2). An improvement in sample size would be necessary to better define the reference intervals for these methods [73].

## Hemoglobin binding protein

Hemoglobin-binding protein was assayed in 184 *T. ornata* from one study site over the course of three years (Table 1, Fig 2). A male bias was present in 2016 and 2017 (p = 0.04 each year).

**Table 2. Marginal effects of sex, physical exam, and packed cell volume on erythrocyte sedimentation rate in free-living eastern (*Terrapene carolina carolina*) and ornate box turtles (*Terrapene ornata ornata*).** Marginal effects were estimated from the most parsimonious multivariable general linear models for each species (S1 & S2 Tables). Reference level for sex is "female" and reference level for physical exam is "normal".

| | *T. carolina* | | | | *T. ornata* | | | |
|---|---|---|---|---|---|---|---|---|
| | Effect Size (mm) | Standard Error | T-Ratio | P-value | Effect Size (mm) | Standard Error | T-Ratio | P-value |
| **Sex (Male)** | | | | | | | | |
| Winpette | -0.28 | 0.24 | -1.18 | 0.2 | -0.68 | 0.2 | -3.39 | 0.001 |
| Winpette Caliper | -0.29 | 0.24 | -1.2 | 0.2 | -0.45 | 0.17 | -2.6 | 0.01 |
| Microhematocrit Tube | -0.55 | 0.34 | -1.62 | 0.1 | -0.62 | 0.2 | -3.03 | 0.003 |
| **Physical Exam (Abnormal)** | | | | | | | | |
| Winpette | 1.89 | 0.42 | 4.49 | <0.0001 | 0.94 | 0.38 | 2.44 | 0.02 |
| Winpette Caliper | 1.84 | 0.42 | 4.35 | <0.0001 | 0.79 | 0.31 | 2.52 | 0.01 |
| Microhematocrit Tube | 2.43 | 0.6 | 4.04 | 0.0001 | 0.99 | 0.39 | 2.56 | 0.01 |
| **Packed Cell Volume (%)** | | | | | | | | |
| Winpette | -0.08 | 0.01 | -5.27 | <0.0001 | -0.067 | 0.03 | -2.22 | 0.03 |
| Winpette Caliper | -0.09 | 0.01 | -6.51 | <0.0001 | -0.073 | 0.03 | -2.55 | 0.01 |
| Microhematocrit Tube | -0.11 | 0.02 | -5.17 | <0.0001 | -0.1 | 0.03 | -3.4 | 0.0009 |

**Table 3. Passing-Bablok agreement analysis parameters comparing three different measurement methods for erythrocyte sedimentation rate in free-living eastern (*Terrapene carolina carolina*) and ornate box turtles (*Terrapene ornata ornata*).**

| Test Method | Reference Method | Kendall's Tau (P-value) | Slope (95% CI) | Y-intercept (95% CI) | Error Present |
|---|---|---|---|---|---|
| **T. carolina** | | | | | |
| Microhematocrit Tube | Winpette Caliper | 0.7 ($< 0.0001$) | 1.39 (1.25, 1.55) | 0.27 (-0.35, 0.65) | Proportional |
| Microhematocrit Tube | Winpette | 0.67 ($<0.0001$) | 1.53 (1.3, 1.75) | 0.17 (-0.5, 0.8) | Proportional |
| Winpette Caliper | Winpette | 0.78 ($< 0.0001$) | 1.1 (1, 1.2) | 0.05 (-0.3, 0.4) | None |
| **T. ornata** | | | | | |
| Microhematocrit Tube | Winpette Caliper | 0.55 ($<0.0001$) | 1.38 (1.17, 1.67) | -0.89 (-1.98, 0.09) | Proportional |
| Microhematocrit Tube | Winpette | 0.43 ($<0.0001$) | 0.77 (0.53, 1) | 0.7 (-4, 1.74) | None |
| Winpette Caliper | Winpette | 0.6 ($< 0.0001$) | 1 (0.7, 1.25) | 0.3 (-0.75, 1.23) | None |

Twenty-two turtles were classified as "unhealthy" due to active shell injuries; one individual also had a draining pedal granuloma, and another had a burn injury on the carapace.

Hemoglobin-binding protein concentrations were not significantly different between years (p = 0.8) or sexes (p = 0.2), but HBP concentrations were significantly increased in adults (effect size = 0.11 mg/mL, p = 0.0005) and unhealthy turtles (effect size = 0.07 mg/mL, p = 0.008). HBP concentrations were also not significantly associated with sample storage time (p = 0.6). Multivariable modeling was not pursued due to the low number of significant predictor variables. Reference intervals were constructed for adult *T. ornata*, while summary data are reported for juveniles in accordance with ASVCP guidelines (Table 5) [73].

## Discussion

Understanding wildlife health is vital for supporting public health, food safety, ecosystem functioning, and conservation initiatives. Effective wildlife health assessment protocols depend on reliable diagnostic tests; and identifying clinically useful assays in wild animals is an important area of active research. The present study investigated two components of the acute phase

**Table 4. Summary data including data distribution, measure of central tendency (mean for normally distributed variables, median for non-normally distributed variables), measure of dispersion (standard deviation for normally distributed variables, 10th– 90th percentiles for non-normally distributed variables), and reference intervals for erythrocyte sedimentation rate in free-living, apparently healthy adult eastern (*Terrapene carolina carolina*) and ornate box turtles (*Terrapene ornata ornata*).** CI = confidence interval.

| Species | Method | N | Distribution | Central Tendency (mm) | Dispersion (mm) | Min (mm) | Max (mm) | Reference Interval (mm) | 90% CI Lower Bound | 90% CI Upper Bound |
|---|---|---|---|---|---|---|---|---|---|---|
| *T. carolina* | Winpette | 83 | Non-normal | 3 | 2–4 | 1 | 5 | 1–5 | 0–1 | 5–5.1 |
| *T. carolina* | Winpette Caliper | 83 | Normal | 3.3 | 1 | 1.2 | 5.6 | 1.7–5.4 | 1.6–2.2 | 5.2–5.8 |
| *T. carolina* | Microhematocrit Tube | 79[b] | Non-normal | 4.6 | 3.1–6.4 | 2.6 | 8.2 | 2.6–8.1 | 2.3–2.6 | 8–8.9 |
| *T. ornata* | Winpette | 93 | Non-normal | 4 | 3–5 | 0 | 7 | 2–6 | 2–4[a] | 5–6[a] |
| *T. ornata* | Winpette Caliper | 73[c] | Normal | 3.8 | 0.79 | 2.2 | 5.7 | 2.5–5.4 | 2.3–2.5 | 5.1–5.8[a] |
| *T. ornata* | Microhematocrit Tube | 92[d] | Normal | 4.4 | 1.02 | 2.5 | 6.9 | 2.7–6.7 | 2.4–2.9 | 6.5–7.2 |

[a] Width of Confidence Interval / Width of Reference Interval > 0.2

[b] Outliers = 1.1 mm, 8.3 mm, 8.3 mm

[c] Outliers = 2 mm, 6.6 mm

[d] Outliers = 2 mm

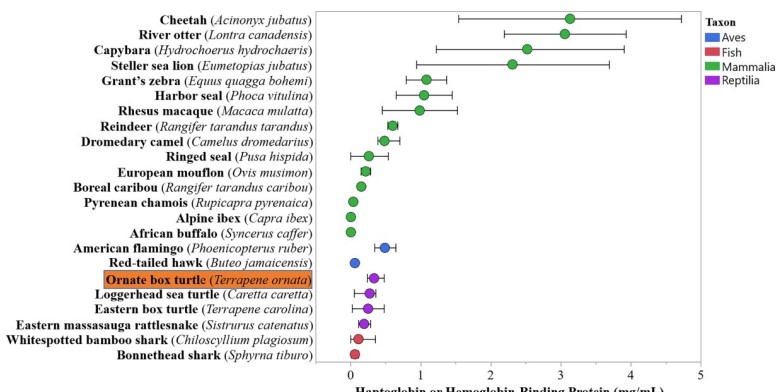

**Fig 2. Comparison of circulating haptoglobin (a.k.a. hemoglobin-binding protein) concentrations in free-living and zoo-maintained wildlife from the present study (ornate box turtles, *Terrapene ornata ornata*) and the literature [9,10,14,15,23,57,58,65–67,86–97].** Circles represent measures of central tendency (mean or median) and bars represent a measure of dispersion (standard deviation or 25th and 7th percentiles).

response in chelonians, a group in which health assessment is inherently complicated by significant physiologic variability. We established reference intervals for hemoglobin-binding protein and erythrocyte sedimentation rate in two species of wild box turtles, characterized several factors (packed cell volume, age class, and sex) which influence test interpretation, and demonstrated that both assays may be useful for detecting acute illness and injury. Our results have direct applications for box turtle and ecosystem health assessment, and should prompt the evaluation of acute phase response testing in other wild reptiles.

Assay performance for both HBP and ESR appears adequate for research and diagnostic purposes in box turtles. In *T. ornata*, HBP concentrations were linearly quantifiable and precise, with an inter-assay CV of 4.23% and an intra-assay CV of 3.82%. These values are comparable to those previously reported for *T. carolina*, (6.6% intra-assay CV) [23] and to those published by the Winpette kit manufacturer (4.1–5.7% inter-assay CV, 5.3–6.3% intra-assay CV). The limit of detection for HBP in *T. ornata* (0.01 mg/mL) was higher than that reported by the manufacturer (0.005 mg/mL), but lower than the cutoff for *T. carolina* (0.08 mg/mL) [23]. Only one turtle had an HBP concentration close to the cutoff (0.092 mg/mL), indicating that this detection limit is likely adequate for quantification of biologically relevant values in *T. ornata*. While assay performance characteristics are not available for the Winpette kit, the 7.9% intra-assay CV that we determined for box turtles is comparable to multiple other clinically acceptable ESR measurement methods in humans [98,99]. Measurements produced by MHT were even more precise than Winpette values, evidenced by a lower intra-assay CV value (3.2%). This indicates that both methods likely have adequate precision for clinical use.

**Table 5. Summary data including data distribution, measure of central tendency (mean for normally distributed variables, median for non-normally distributed variables), measure of dispersion (standard deviation for normally distributed variables, 10th– 90th percentiles for non-normally distributed variables), and reference intervals for hemoglobin-binding protein in free-living, apparently healthy ornate box turtles (*Terrapene ornata ornata*).** CI = confidence interval.

| Age | N | Distribution | Central Tendency (mg/mL) | Dispersion (mg/mL) | Min (mg/mL) | Max (mg/mL) | Reference Interval (mg/mL) | 90% CI Lower Bound | 90% CI Upper Bound |
|---|---|---|---|---|---|---|---|---|---|
| Adult | 145[a] | NG | 0.353 | 0.27–0.505 | 0.162 | 0.667 | 0.198–0.606 | 0.167–0.217 | 0.567–0.654 |
| Juvenile | 13 | G | 0.258 | 0.09 | 0.162 | 0.472 | NA | NA | NA |

[a] Outliers = 0.092 mg/mL, 0.157 mg/mL, 0.892 mg/mL, 0.904 mg/mL

Box turtle erythrocyte sedimentation rate and hemoglobin-binding protein values were generally consistent with previously-published reptilian results and followed demographic, physiologic, and health patterns identified in other taxa (Figs 1 and 2). Like humans and Indian softshell turtles (*Nilssonia gangetica*), box turtle ESR was negatively associated with packed cell volume. In humans this relationship is due to changes in frictional forces based on erythrocyte concentration, and while the same mechanism likely explains this finding in other species, the exact cause has not been determined in reptiles [45,46,61]. Female *T. ornata*, which were likely sampled during periods of vitellogenesis, had greater ESR values than males. A similar trend has also been documented in humans, oriental garden lizards (*Calotes versicolor*), Bengal monitor lizards (*Varanus bengalensis*), Indian softshell turtles, beluga whales (*Delphinapterus leucas*), and ostriches (*Struthio camelus*) [59–64,100]. Some authors postulate that this is due to differences in erythrocyte size and number between the sexes, but an exact mechanism is not readily agreed upon [61]. In reptiles, additional explanations could include changes in plasma protein, lipoprotein, lipid, and immunoglobulin concentrations during vitellogenesis [18,19]. Sex was not a significant predictor of ESR in *T. carolina*, however, this species was sampled at multiple points during the reproductive cycle. A sex effect may emerge with focused evaluation during periods of peak reproductive activity. Advancing age is associated with greater ESR values in humans, mute swans (*Cygnus olor*), Atlantic bottlenose dolphins (*Tursiops truncatus*), and Bengal monitors, while the opposite is observed in ostriches, beluga whales, and killer whales (*Orcinus orca*) [45,49,56,59,62,64,101]. Juvenile sample size (two *T. carolina*, four *T. ornata*) was inadequate to vigorously assess the effects of age on ESR in the present study, though age-based differences may be identified with a larger sample size. Future studies should target *T. carolina* during vitellogenesis, include larger numbers of juveniles, and incorporate data from multiple years to better determine the effects of age, sex, and temporal variation on ESR.

Health status was also a statistically significant predictor of ESR values in box turtles. The magnitude of ESR elevation in unhealthy turtles was small (2 mm in *T. carolina*, 1 mm in *T. ornata*) but consistent with previous findings in gopher tortoises (median value for healthy tortoises = 5 mm, median value for unhealthy tortoises = 7 mm) [25,48]. In *G. polyphemus*, an ESR cutoff of 5 mm was 79% sensitive and 75% specific for differentiating healthy and unhealthy tortoises, providing better discriminatory capability than leukocyte counts, lactate, fibrinogen, and protein electrophoresis fractions [48]. The limited sample size in the present study (six "unhealthy" *T. carolina*, eight "unhealthy" *T. ornata*) precluded establishment of reliable cutoff values. However, applying the 5 mm cutoff from gopher tortoises to our dataset produces comparable results: *T. carolina* sensitivity = 67% and specificity = 93%, *T. ornata* sensitivity = 75% and specificity = 74%. This suggests that box turtle ESR values may still have clinical relevance despite the absence of large differences between healthy and unhealthy animals typical of mammals (e.g. 13–111 mm) [52,102–108]. Additional studies are needed to determine optimum cutoff values for ESR in box turtles and to assess the diagnostic performance of this test in *T. carolina* and *T. ornata*.

Hemoglobin-binding protein concentrations were significantly increased in adult *T. ornata* compared to juveniles. Aging has also been associated with increased HBP concentration in Steller sea lions (*Eumetopias jubatus*), ringed seals (*Pusa hispida*), and *T. carolina*, likely due to differences in pathogen burden and exposure to inflammatory stimuli between age classes [10,23,57,58]. Hemoglobin-binding protein concentrations are increased in male ringed seals and rhesus macaques (*Macaca mulatta*) compared to females, while the reverse is true for *T. carolina* [23,58,66]. In the present study, HBP concentrations from *T. ornata* did not differ between sexes, however, all turtles were evaluated shortly after waking from brumation in early spring. It is possible that reproductive differences would emerge if turtles were sampled

at more time points during the active season. Seasonal differences in HBP concentrations have been identified in Pyrenean chamois (*Rubricapra pyrenaica pyrenaica*) and site differences have been detected in Steller sea lions, indicating the potential need for season and population-specific reference intervals [10,91,94]. All HBP measurements in the present study came from a single box turtle population that was serially sampled over the course of three years, and the potential for seasonal and population-level variability in this analyte is unknown. Future studies on HBP in *T. ornata* should include samples from different seasons and populations to determine the importance of spatiotemporal factors for clinical interpretation of this analyte.

Unhealthy *T. ornata* had a 1.2-fold increase in HBP concentrations, which would classify HBP as a minor acute phase protein in this species [2]. However, it is important to consider that these "unhealthy" turtles were heterogeneous in terms of clinical presentation and duration of clinical signs. Several studies have demonstrated that the magnitude of HBP change in unhealthy animals depends on the underlying inflammatory stimulus. For example, capybara (*Hydrochoerus hydrochaeris*) naturally infected with sarcoptic mange (*Sarcoptes scabiei*) had an almost five-fold increase in plasma haptoglobin concentrations compared to uninfected individuals [65]. However, capybara injected with turpentine to stimulate an inflammatory response only developed a 2.7-fold increase in plasma haptoglobin concentrations [65]. African buffalo (*Syncerus caffer*) developed significantly increased plasma haptoglobin concentrations after experimental challenge with foot-and-mouth disease virus and natural exposure to parainfluenza virus and *Mycobacterium bovis*, but not following natural exposure to five other bovine pathogens [15]. In Steller sea lions, plasma and serum haptoglobin concentrations increase in response to temporary captivity, hot-branding, and subcutaneous abscessation, but not following blubber biopsy [109,110]. Rhesus macaques experimentally infected with *Mycobacterium tuberculosis* had serum haptoglobin concentrations 2-fold higher then uninfected individuals, but haptoglobin concentrations did not significantly change for individuals with acute traumatic injuries (e.g. bite wounds) [66]. In dromedary camels (*Camelus dromedarius*), serum haptoglobin concentrations increased approximately 4-fold following surgical castration, while chemical contraception via gonadotropin releasing hormone vaccination resulted in an approximately 2.4-fold increase [92]. In contrast, disseminated lymph node abscessation secondary to *Corynebacterium* sp. infection was only associated with a 1.4-fold increase in serum haptoglobin concentrations [92]. These studies demonstrate that increases in haptoglobin concentration may be diluted if animals with multiple different inflammatory stimuli are considered together, which may partially explain the small effect size observed for unhealthy *T. ornata* in the present study.

The ability to detect changes in HBP concentrations also depends on the timing of sample collection relative to insult. Capybara injected with turpentine had increased plasma haptoglobin concentrations from one to three weeks post-injection, with peak values reached two weeks post-injection [65]. African buffalo challenged with foot-and-mouth disease virus achieved peak plasma haptoglobin concentrations five days following inoculation and values remained elevated above baseline for 21 days following viral clearance [15]. In hot-branded Steller sea lions, serum haptoglobin concentrations peaked one week following branding and returned to baseline levels by eight weeks post-branding [110]. The inclusion of box turtles with different durations of illness or injury may have reduced the effect of health status on plasma HBP concentrations in the present study. Indeed, it is not uncommon to detect small differences in haptoglobin concentration between apparently healthy and unhealthy animals in cross-sectional studies (e.g. 1.4–2.6-fold), likely due to heterogeneity in etiology and length of illness [14,67,90,92,97]. Despite this, relatively small changes in haptoglobin concentrations can still represent biologically important differences in health status. For example, haptoglobin

concentrations were only increased 1.6–1.75-fold in declining populations of Alaskan pinnipeds compared to historically stable populations [10] and were only 1.2-fold higher in oiled populations of river otters (*Lontra canadensis*) compared to animals at non-oiled sites [9]. The detection of increased plasma HBP concentrations in sick and injured *T. ornata* is consistent with findings in other wildlife studies, and may support the use of HBP in box turtle health assessment. Additional studies are needed to determine which disease processes and inflammatory stimuli produce the greatest increase in HBP concentration, and to determine the time course of HBP elevation in unhealthy *T. ornata*.

ESR and HBP have several diagnostically attractive features in box turtles including consistency between seasons (ESR), populations (ESR), and years (HBP). Clinical analytes with minimal demographic and physiologic variability are appealing because extreme values are more likely to represent pathologic change [14]. Furthermore, diagnostic tests which sensitively detect a variety of health problems can enhance the efficiency of wildlife health surveillance [14]. Including ESR and HBP in box turtle health assessment protocols may therefore facilitate the identification of unhealthy animals and promote the application of targeted management strategies. Further studies are needed to assess the magnitude and duration of ESR and HBP changes secondary to a variety of inflammatory stimuli, and to determine the optimal method for ESR measurement.

Erythrocyte sedimentation rate values can vary by both anticoagulant and measurement method. ESR is typically performed using blood anticoagulated with EDTA or sodium citrate [46]. The present study performed ESR using heparinized whole blood samples due to the potential for unpredictable hemolysis using EDTA [111,112]. While other reptilian ESR studies have also used heparinized samples, it is important to recognize that results may vary with different anticoagulants [25,48,83]. Winpettes rely on the Wintrobe method for ESR determination, which measures undiluted, anticoagulated blood samples in a 100mm tube. While Wintrobe ESR has been performed for decades, the International Council for Standardization in Haematology considers the Westergren method to be the gold standard for ESR determination in humans due to improved sensitivity and high reproducibility. The Westergren method was not utilized in this study due to the larger blood volume requirement (2mL), which would make measurement of ESR in juvenile box turtles impossible and would severely limit the number of additional blood tests that could be performed for each adult. The results of Wintrobe and Westergren ESR in humans do not always reliably correlate, therefore the results of the present study may not correspond to ESR values determined using different methods.

Hemoglobin-binding protein was determined in batches and some plasma samples were frozen for up to two years before analysis. Previous studies have shown that HBP remains stable in stored serum samples for up to 4 years, and the plasma storage methods utilized in the present study were likely adequate to preserve HBP activity [113]. An additional potential limitation in this study is the use of the subcarapacial sinus for venipuncture. The risk of lymphatic contamination is considered higher at this site than from an isolated vessel such as the jugular vein [114,115]. However, jugular venipuncture is time-consuming, requires multiple people for restraint and phlebotomy, and entails full extension of the neck and restraint of the forelimbs, which can be stressful for the turtle. Venipuncture from the subcarapacial sinus can be performed rapidly by a single phlebotomist without opening the turtle's shell, and is therefore more practical to perform when processing large numbers of turtles in a field setting. Diagnostic tests including hematology and biochemistry panels could have been pursued to evaluate for characteristic changes associated with lymphatic contamination, but this was beyond the scope of the study. While obviously lymph-contaminated samples were excluded, inapparent lymph contamination can never be fully ruled out, and increased variability in ESR and HBP may have resulted.

## Conclusions

This study provides initial characterization of two promising diagnostic tests in free-living box turtles. Health status and packed cell volume influenced ESR in both species, while sex was also an important driver in *T. ornata*. Hemoglobin-binding protein concentrations were affected by health status and age class. Erythrocyte sedimentation rate values were similar between seasons and populations, and HBP concentrations were consistent between years, indicating that these analytes may have more stable baseline values than traditional health metrics in reptiles. We also demonstrated that performing ESR in a microhematocrit tube may be a viable alternative to commercial kits if test-specific reference intervals are utilized. The reference intervals generated in this study represent the first steps towards clinical application of ESR and HBP in free-living box turtles. Continued exploration of novel health assessment tools may ultimately improve veterinary understanding of reptilian health and disease, resulting in better conservation outcomes and more precise monitoring of ecosystem wellness.

## Supporting information

**S1 File. All data analyzed in this study.**
(XLSX)

**S1 Table. Model selection parameters for general linear models predicting erythrocyte sedimentation rate in free-living eastern box turtles (*Terrapene carolina carolina*).** N = sample size, K = number of parameters estimated for each model, $AIC_c$ = Akaike's information criterion corrected for sample size, $\Delta AIC_c$ = Difference in Akaike's information criterion compared to the most parsimonious model, $w_i$ = Akaike weight.
(DOCX)

**S2 Table. Model selection parameters for general linear models predicting erythrocyte sedimentation rate in free-living ornate box turtles (*Terrapene ornata ornata*).** N = sample size, K = number of parameters estimated for each model, $AIC_c$ = Akaike's information criterion corrected for sample size, $\Delta AIC_c$ = Difference in Akaike's information criterion compared to the most parsimonious model, $w_i$ = Akaike weight.
(DOCX)

**S1 Fig. Bland-Altman plots comparing three different measurement methodologies for Erythrocyte Sedimentation Rate (ESR) in free-living eastern (*Terrapene carolina carolina*) and ornate box turtles (*Terrapene ornata ornata*).** Top row: *T. carolina*, Bottom row: *T. ornata*. Central dashed line = Mean difference between measurement methodologies, Top and bottom dashed lines = limits of agreement, defined as the mean difference +/- 1.96 times the standard deviation of the differences. MHT = microhematocrit tube ESR.
(TIF)

**S2 Fig. Passing-Bablok regression plots comparing three different measurement methodologies for Erythrocyte Sedimentation Rate (ESR) in free-living eastern (*Terrapene carolina carolina*) and ornate box turtles (*Terrapene ornata ornata*).** Top row: *T. carolina*, Bottom row: *T. ornata*. Dashed line = line of perfect agreement with slope = 1 and y-intercept = 0. Solid line: Passing-Bablok regression line. Shaded region: 95% confidence interval of Passing-Bablok regression line. MHT = microhematocrit tube ESR.
(TIF)

## Acknowledgments

We thank John Rucker and the turtle dogs for their assistance with locating turtles, the Turtle Team veterinary students for their assistance with sampling, and Bayer for providing flea and tick preventative for the dogs.

## Author Contributions

**Conceptualization:** Laura Adamovicz.

**Data curation:** Laura Adamovicz, Sarah J. Baker, Ethan Kessler, Marta Kelly, Samantha Johnson, John Winter, Christopher A. Phillips, Matthew C. Allender.

**Formal analysis:** Laura Adamovicz.

**Funding acquisition:** Laura Adamovicz, Matthew C. Allender.

**Investigation:** Laura Adamovicz.

**Methodology:** Laura Adamovicz.

**Writing – original draft:** Laura Adamovicz, Sarah J. Baker, Ethan Kessler, Marta Kelly, Samantha Johnson, John Winter, Christopher A. Phillips, Matthew C. Allender.

**Writing – review & editing:** Laura Adamovicz, Sarah J. Baker, Ethan Kessler, Christopher A. Phillips, Matthew C. Allender.

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
