## [Decision Letter · Decision Letter 0]

16 Apr 2020

PONE-D-20-02328

Novel diagnostics facilitate health assessment in free-living box turtles (Terrapene spp.)

PLOS ONE

Dear Dr Adamovicz,

Thank you for submitting your manuscript to PLOS ONE. After careful consideration, we feel that it has merit but does not fully meet PLOS ONE’s publication criteria as it currently stands. Therefore, we invite you to submit a revised version of the manuscript that addresses the points raised during the review process.

While both reviewers agree that there is valuable information presented in your manuscript. Reviewer 1 offers many comments for improvement and clarification that should be addressed.

We would appreciate receiving your revised manuscript by May 31 2020 11:59PM. To enhance the reproducibility of your results, we recommend that if applicable you deposit your laboratory protocols in protocols.io, where a protocol can be assigned its own identifier (DOI) such that it can be cited independently in the future. For instructions see: http://journals.plos.org/plosone/s/submission-guidelines#loc-laboratory-protocols

We look forward to receiving your revised manuscript.

Kind regards,

Ulrike Gertrud Munderloh, Ph.D.

Academic Editor

PLOS ONE

Reviewers' comments:

Reviewer's Responses to Questions

**Comments to the Author**

1. Is the manuscript technically sound, and do the data support the conclusions?

Reviewer #1: Yes

Reviewer #2: Yes

2. Has the statistical analysis been performed appropriately and rigorously? 

Reviewer #1: Yes

Reviewer #2: Yes

3. Have the authors made all data underlying the findings in their manuscript fully available?

Reviewer #1: Yes

Reviewer #2: No

4. Is the manuscript presented in an intelligible fashion and written in standard English?

Reviewer #1: Yes

Reviewer #2: Yes

5. Review Comments to the Author

Reviewer #1: Title: Many of these are not new diagnostics, they a quite old, just new for this species. I addition as written the title is overly vague. Suggested title revision: Comparison Acute Phase Reactants for Health Assessment of free-living box turtles (Terrapene spp,)

Throughout the manuscript, sentences should not begin with acronyms. I count 13 uses of acronyms in the abstract; reduce the use of acronyms in the abstract to only those absolutely necessary. I recommend removal of APR, OBT/EBT (use shortened scientific name instead) and using acronyms only for tests to increased abstract readability.

Line: 209, 239, 344, 370, 123, 268, 279, 331382,

236 provides an example of oververbosity which can and should be summarized throughout the MS. Please review the entire text for these instances: Results for the single juvenile EBT were 4, 4.1, and 6 mm/hr for the Winpette, caliper measured Winpette and the Microhematocrit methods, respectively.

Line 102 - provide field sites in additional files

Line 109 without clinical signs

Line 115 - provide full details was pain control provided

Line 113-115 - which test was determined from which, separator gels can interfere with some assays

Line 128 -Provide full details, make manufacturer

Line 141 - The differences in time of sample storage and results should be assessed statistically

Line 134- provide full details, make manufacture of "stand"

Line 167 - provide full details, AICmodavg package

Line 177 - ReferenceIntervals package: provide full details

Line 346 - You have not proven this for box turtles

Line 292- 294 Inter-assay variation is generally provided as the beginning of the results and reviewed justified in the discussion as acceptable or unacceptable based on previous studies. further discussion appears lacking.

Line 355 - you didn't vary your anticoagulant in this study...

In the goals of the study and with the discussion you state you " Characterized the spatial, temporal, demographic, and physiologic factors which influenced test interpretation". this statement both within goals and discussion is overly broad, you characterized selected, or some factors: In the discussion and the introduction I would specify which you factors tested and which category you believe they belong to.

Additional points for the discussion: Justify your choice of the SCC considering: Jugular vein is known to have less lymphatic contamination and no CSF contamination. Justify determination method of determination of lack of lymphatic contamination or other ways this could have been determined. Was hemolysis or lipemia assessed in the study and if so or if not how might this have affected results, shouldbe in discussion

Line 120: it appears that samples were moved from a heparinized tube into yet another heparinized to for HCT determination, increasing the likelihood of hemolysis, was this assessed?

311: You suggest this may be due to erythrocyte frictional forces as in other species but do not know this to be true in these current study. Other factors which also change based on sex a reproduction could also change ESR: increased proteins, lipoproteins and immunoglobulins of vittellogensis could affect ESR.

Figs: ESR and Haptoglobin - Provide scientific names as well as common names. Consider additional figures of ectotherms only (or reptiles only, preferred) with box plot to better detail difference among species more closely taxonomically related. Some reference are cited within the figure, others are not. I suggest provide all reference citation at the end of the Figure titles for these comparative figures.

Tables have excessive acronyms. EBT, OBT within the table should be replaced with the scientific name OR the common name with the scientific name give in the Table info

Table 2 NEW TITLE: Effect of Free-Living Box Turtle Sex, Physical Examination, and Packed Cell Volume upon Erythrocyte Sedimentation Rates based on multivariable general linear models. Winpette, Winnpette C and ( MicroHCT should be expanded within the table. Se, T, The test for the P value and Effect size should be defined. Are the miniscule changes found to be statistically different here really clinically revelant, what constitutes clinical relevancy in other species for these tests? This should be addressed in the discussion

Table 3: You have plant of space within the table to spell out and provide the scintific name of each species

A less shortened form of each method could also be used within the table. Last Column Header Proportional error, results Yes or No below.

Table 4 : Title should be revised into summary form, there is much redundancy. Method and Age should be different columns. Suggest separate tables for HBP and ESR as there is no benefit in placing them together.

ESR is measured as a rate and is therefore either faster or slower than OR relatively increased or decreased in rate compared to other measurements. It is not higher (elevated) or lower. Please correct throughout the manuscript.

HBP is measured as a concentration and is therefore has a relatively increased or decreased concentration in comparison to other concentrations; it is not higher (elevated) or lower. Please correct throughout the manuscript.

Tables S1 & S2: Model attributes are not defined

I hope these comments improve this manuscript, which provides some great information :)

Reviewer #2: This manuscript is technically sound and I have very few specific or general comments. This study should be helpful to reptile physiological ecologists interested in pursuing field-based studies in non-model species (which is a never-ending problem to resolve when many diagnostic assays are species specific and/or reagent limited).

My only two editorial comments are below:

Line 94-95: I'm not sure for the rationale behind expectations and predictions, especially that of females. Please describe more context behind the specific hypotheses in this study, so that this study reads as more than simply being a technological question of comparing multiple assays in free-living turtles.

Linen 284: “data” is a plural word, thus this should state “summary data are”

6. PLOS authors have the option to publish the peer review history of their article (what does this mean?). If published, this will include your full peer review and any attached files.

Reviewer #1: No

Reviewer #2: No

---

## [Author Response · Author response to Decision Letter 0]

31 May 2020

Please see the attached rebuttal letter which provides a detailed response to each question/comment from the reviewers

---

## [Editor Report · Decision Letter 1]

3 Jun 2020

Erythrocyte sedimentation rate and hemoglobin-binding protein in free-living box turtles (Terrapene spp.)

PONE-D-20-02328R1

Dear Dr. Adamovicz,

We’re pleased to inform you that your manuscript has been judged scientifically suitable for publication and will be formally accepted for publication once it meets all outstanding technical requirements.

Kind regards,

Ulrike Gertrud Munderloh, Ph.D.

Academic Editor

PLOS ONE
---

## [Editor Report · Acceptance letter]

8 Jun 2020

PONE-D-20-02328R1 

Erythrocyte sedimentation rate and hemoglobin-binding protein in free-living box turtles (*Terrapene* spp.) 

Dear Dr. Adamovicz:

I'm pleased to inform you that your manuscript has been deemed suitable for publication in PLOS ONE. Congratulations! Your manuscript is now with our production department. 

Kind regards, 

on behalf of

Dr. Ulrike Gertrud Munderloh 

Academic Editor

PLOS ONE